# Latent Feature Group Learning for High-Dimensional Data Clustering

**Wenting Wang** [1,2] , **Yulin He** [1,2,*] **, Liheng Ma** [3] **and Joshua Zhexue Huang** [1,2]

1   Big Data Institute, College of Computer Science and Software Engineering, Shenzhen University, Shenzhen 518060, China; wangwt@szu.edu.cn (W.W.); zx.huang@szu.edu.cn (J.Z.H.)
2   National Engineering Laboratory for Big Data System Computing Technology, Shenzhen University, Shenzhen 518060, China
3   School of Computer Science, McGill University, Montreal, QC H3A OG4, Canada; liheng.ma@mail.mcgill.ca
*   Correspondence: yulinhe@szu.edu.cn

**Abstract:** In this paper, we propose a latent feature group learning (LFGL) algorithm to discover the feature grouping structures and subspace clusters for high-dimensional data. The feature grouping structures, which are learned in an analytical way, can enhance the accuracy and efficiency of high-dimensional data clustering. In LFGL algorithm, the Darwinian evolutionary process is used to explore the optimal feature grouping structures, which are coded as chromosomes in the genetic algorithm. The feature grouping weighting *k*-means algorithm is used as the fitness function to evaluate the chromosomes or feature grouping structures in each generation of evolution. To better handle the diverse densities of clusters in high-dimensional data, the original feature grouping weighting *k*-means is revised with the mass-based dissimilarity measure rather than the Euclidean distance measure and the feature weights are optimized as a nonnegative matrix factorization problem under the orthogonal constraint of feature weight matrix. The genetic operations of mutation and crossover are used to generate the new chromosomes for next generation. In comparison with the well-known clustering algorithms, LFGL algorithm produced encouraging experimental results on real world datasets, which demonstrated the better performance of LFGL when clustering high-dimensional data.

**Keywords:** subspace clustering; feature grouping; genetic algorithm; high-dimensional data analysis; evolutionary computing

---

## 1. Introduction

High-dimensional data analysis is a challenging task in the domains of machine learning and artificial intelligence. Thousands of features in high-dimensional data cause a high complexity when using the existing tools for low-dimensional data to cluster high-dimensional data [1]. High-dimensional data often contain many redundant, irrelevant and noise features, which affect the clustering results. In the past decades, many subspace clustering algorithms have been proposed to handle high-dimensional data, aiming at finding clusters from subspaces of data rather than the entire data space [2]. Among the various subspace clustering methods, soft subspace clustering is an important technique. It assigns the weights to individual features and uses the weights to identify important features where the subspace structures of clusters can be discovered [3,4].

The individual feature weighting method suffers from the high-dimensionality of data with hundreds of thousands of features. The first problem is that the feature weights produced by the individual feature weighting method are unstable. For example, experimental results reveal that the individual feature weights can converge to similar values in low-dimensional data [5]. However, with the increase of data dimensionality, the feature weights calculated with different initial values

become unstable and no longer converge [6]. The second problem is that many weak features exist in high-dimensional data so it is hard to recover the subspace cluster structure from the data. The third problem is that the clusters are often buried in many noise features.

To tackle the above-mentioned problems, the feature grouping weighting *k*-means algorithm named FG-*k*-means [7] is proposed for high-dimensional data. In FG-*k*-means, the features are divided into a small set of feature groups, where each being treated as a group feature in the low dimensional space of feature groups. High-dimensional data that are clustered on the group features and the clusters in different subspaces of group features are discovered by assigning the weights to group features [6]. Because the group features generalize the information of individual features in high-dimensional data, the FG-*k*-means algorithm performs better than the clustering algorithms, which cluster the data on the individual features. One limitation of FG-*k*-means is that the feature groups have to be known in advance. However, very few high-dimensional datasets in the real world have the feature groups available. This requirement limits the practical use of FG-*k*-means algorithm.

In this paper, we propose a latent feature group learning (LFGL) algorithm to automatically learn the latent feature groups in the process of subspace clustering for high-dimensional data. This algorithm consists of two levels of optimizations. The outer level of optimization uses the Darwinian evolutionary process to learn the optimal structure of feature groups. In this process, the feature group structures are coded as the chromosomes in genetic algorithm and the best chromosome is selected through evolutions of generations. The inner level of optimization is used in each generation to evaluate the chromosomes and select the stronger ones for genetic operations to generate the new chromosomes in the next generation of the Darwinian evolution process. The FG-*k*-means algorithm is used in the inner level of optimization as the fitness function for chromosome selection. To effectively deal with complex high-dimensional data, two revisions are made in the FG-*k*-means algorithm. The mass-based dissimilarity measure is used to replace the Euclidean distance in calculating the dissimilarity between objects so the different densities of clusters can be better handled. The feature grouping and group weights are optimized with a convex penalty relaxation method by using the orthogonal constraint to ensure the orthogonality among the feature groups. The relaxation problem is solved with the process of nonnegative matrix factorization. At the outer level of optimization, mutation and crossover operations are used to manipulate chromosomes for the next generation. We conducted experiments on six gene datasets and one text dataset. We compared the results of LFGL algorithm with five existing algorithms. Encouraging experimental results were obtained by LFGL algorithm, which demonstrated the better performance of proposed LFGL algorithm.

The remainder of this paper is organized as follows. In Section 2, we review some related work. In Section 3, we present the latent feature grouping model for projection of high-dimensional data to a low-dimensional space. Section 4 presents the LFGL algorithm. The experimental results are presented and discussed in Section 5. The conclusions are drawn in Section 6.

## 2. Related Work

In the past decade, the soft subspace clustering has been an important research topic in cluster analysis [4–21]. The representative works are summarized as follows.

- Huang et al. [5] proposed W-*k*-means clustering algorithm which can automatically compute the feature weights in *k*-means clustering process. W-*k*-means extends the standard *k*-means algorithm with one additional step to compute the feature weights at each iteration of clustering process. The feature weight is inversely proportional to the sum of within-cluster variances of feature. As such, noise features can be identified and their effects on the clustering result are significantly reduced. Amorim et al. [22] extended W-*k*-means algorithm with Minkowski's metric and assigned the Minkowski's exponent that coincides with the exponent to feature weights.
- Hoff [17] proposed a multivariate Dirichlet process mixture model, which is based on a Pólya urn cluster model for the multivariate means and variances. The model is learned by a Markov chain Monte Carlo process. However, its computational cost is prohibitive. Fan et al. [23] proposed

a variational inference framework for unsupervised non-Gaussian feature selection, in the context of the finite generalized Dirichlet (GD) mixture-based clustering. Under the proposed principled variational framework, it simultaneously estimates all the involved parameters and determines the complexity (i.e., both model and feature selection) of GD mixture.

- Domeniconi et al. [16] proposed the locally adaptive clustering (LAC) algorithm, which assigns a weight for each feature in the cluster. They used an iterative algorithm to minimize the objective function. Cheng et al. [21] proposed another weighting *k*-means approach very similar to LAC, but allowing for incorporation of further constraints. Jing et al. [4] proposed the entropy weighting *K*-means (EWKM) algorithm, which also assigns a weight to each feature in each cluster. Different from LAC, EWKM extends the standard *k*-means algorithm with one additional step to compute the feature weights for each cluster at each iteration. The weight is inversely proportional to the sum of within-cluster variances of feature in the cluster.

- Tsai and Chiu [19] developed a feature weights self-adjustment mechanism for *k*-means clustering on the relational datasets, in which the feature weights are automatically computed by minimizing the within-cluster discrepancy and maximizing the between-cluster discrepancy simultaneously. Deng et al. [20] proposed an enhanced soft subspace clustering algorithm (ESSC) which employs both within-cluster and between-cluster information in the subspace clustering process. Xia et al. [24] proposed a multi-objective-evolutionary-approach-based soft subspace clustering (MOEASSC) algorithm which optimizes two minimization objective functions simultaneously by using a multi-objective evolutionary approach. They also proposed a two-step method to reduce the difficulty in identifying the subspace of each cluster, the cluster memberships of objects and the number of clusters.

- Chen et al. [7] proposed a two-level weighting method named FG-*k*-means for multi-view data in which two types of weights are employed: a view weight is assigned to each view to identify the view compactness and a feature weight is also assigned to each feature in a view to identify the feature contribution. Cai et al. [25] used FG-*k*-means for text clustering. They first used the topic model LDA to partition the words into several groups and then used FG-*k*-means to cluster the text data. The experimental results show that the word grouping method improves the clustering performance on text data. Gan et al. [26] extended FG-*k*-means to generate the feature groups automatically by clustering the features weights. The objective function of clustering feature weights is added as a penalty term to the objective function of FG-*k*-means. This indicates that there is a much higher chance to construct all informative abstract features than all informative individual features.

## 3. Latent Feature Group Projection in Subspace Clustering

Although many features are used to describe data in the high-dimensional space, few of them are needed to distinguish a specific cluster from others. Thus, the subspace clustering algorithms [4,5,16,24] obtain better performance than that of general *k*-means algorithm. As the number of dimension increases, the strategy of searching subspace of decisive features in the entire space often leads to the suboptimal results because of the existence of many noise features [6]. Meanwhile, it deteriorates the performance of subspace clustering.

To solve the above-mentioned problem, we project the high-dimensional objects into the low-dimensional latent feature group space. The features in the high-dimensional spaces are not fully independent. Rather, they gather together into nearly mutually exclusive groups. However, the feature groups are unknown in most real world datasets and they are latent in high-dimensional data. To make the projection possible, we need to design an effective learning process that can automatically find the latent feature groups and cluster the objects in the subspace of feature groups. We use two steps to solve this problem. The first step is to build a latent feature grouping model that enables to express the feature partition. The second step is to embed the latent feature grouping model into the

objective function of subspace clustering process, so that grouping features and clustering data can be performed simultaneously.

Figure 1 illustrates the latent feature grouping model. The left column shows the features in a dataset $X_{i,j} \in \mathbb{R}^{n \times m}$. The feature set $\mathcal{A} = \{x_1, x_2, \cdots, x_m\}$ are mapped into $t$ groups $\{g_1, g_2, \cdots, g_t\}$ in the middle column. We can see that each feature is mapped to only one group, which ensures the exclusiveness of features in different groups. The mapping can be defined in a partition matrix $V_0$ in which the row indicates the feature groups and the column indicates the features. If a feature $i$ is mapped to the feature group $j$, the element $(i, j)$ is assigned to 1. Otherwise, it is assigned to 0. Because of the exclusiveness, each column in $V_0$ has only one element with value 1. The rest are 0s. When a feature $i$ is mapped to the feature group $j$, we also assign a weight to the feature to indicate the feature importance in the group. The feature weights are represented in the weight matrix $V_l$, where $l$ is a cluster. We let $V_l = V_0 \circ V_l$. The feature groups are also weighted with $W \in \mathbb{R}_+^t$ to result in the weighted group values $\{g(x)_1, g(x)_2, \cdots, g(x)_t\}$ shown in the right column. The weights of feature groups identify the importance of the feature groups in determining each cluster. Formally, we can write the latent feature grouping model as

$$g(x) = W_l(V_0 \circ V_l)x^T. \tag{1}$$

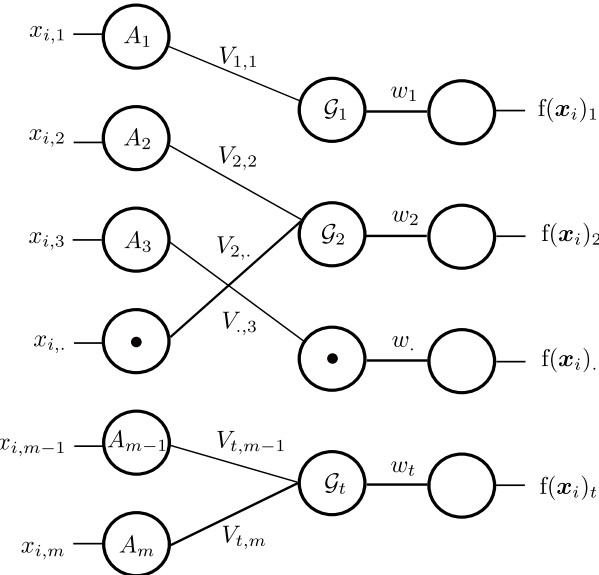

**Figure 1.** Latent feature grouping model.

Below, we use a simple example to illustrate Equation (1). The matrix

$$W_l = \begin{pmatrix} 0.1 & 0 & 0 \\ 0 & 0.2 & 0 \\ 0 & 0 & 0.7 \end{pmatrix} \tag{2}$$

provides the weights of feature groups in cluster $l$ with the normalized constraint $\sum w_l = 1$. A partition of six features into three groups is specified in the partition matrix $V_0$ as

$$V_0 = \begin{array}{c} \\ \mathcal{G}_1 \\ \mathcal{G}_2 \\ \mathcal{G}_3 \end{array} \begin{array}{c} \begin{matrix} A_1 & A_2 & A_3 & A_4 & A_5 & A_6 \end{matrix} \\ \begin{pmatrix} 1 & 1 & 0 & 0 & 0 & 0 \\ 0 & 0 & 0 & 1 & 0 & 1 \\ 0 & 0 & 1 & 0 & 1 & 0 \end{pmatrix} \end{array}, \tag{3}$$

where the rows of $V_0$ represent the feature groups and the element 1 in each column indicates the feature group to which the feature is assigned. The features in each group are also weighted on the importance of each feature in the feature group of each cluster. The weighting scheme for cluster $l$ is represented as the following weight matrix

$$
V_l = \begin{array}{c} \\ \mathcal{G}_1 \\ \mathcal{G}_2 \\ \mathcal{G}_3 \end{array} \begin{array}{cccccc} A_1 & A_2 & A_3 & A_4 & A_5 & A_6 \\ \begin{pmatrix} 0.89 & 0.46 & 0 & 0 & 0 & 0 \\ 0 & 0 & 0 & 0.60 & 0 & 0.80 \\ 0 & 0 & 0.35 & 0 & 0.94 & 0 \end{pmatrix} \end{array}. \tag{4}
$$

The none zero elements in $V_l$ are the same as those in $V_0$. $V_l$s are initialized and optimized on the constraint of $V_l V_l^T = I$, thus we have

$$
\sum_{j=1}^{6} v_{i,j}^2 = 1, \text{ for } 1 \leq i \leq 3. \tag{5}
$$

Since $V_l = V_0 \circ V_l$, the feature group structure $V_0$ and the individual feature weights $V_l$ in each cluster can be optimized separately. In the current feature grouping weighting algorithms such as FG-$k$-means, $V_0$ is supposed to be known in advance and $V_l$s are optimized in the clustering process. However, $V_0$ is usually unknown. Therefore, the current algorithms are not able to learn the feature grouping structure automatically. With the latent feature grouping model in Equation (1), we are able to learn $V_0$ in the latent feature group learning algorithm described in the next section.

## 4. Latent Feature Group Learning Algorithm

In this section, we introduce the latent feature group learning (LFGL) algorithm which can search the optimal latent feature groups and computes the group weights and individual feature weights for each cluster. This is automatically achieved in two levels of optimization. The outer level uses the Darwinian evolution process to select the optimal feature grouping structure $V_0$. The inner level uses the revised FG-$k$-means algorithm to evaluate each feature grouping structure. In the following, we first present the revised FG-$k$-means. Then, we present the Darwinian evolution process for selecting the optimal feature grouping structure.

### 4.1. Revised FG-k-Means

In LFGL algorithm, each given feature grouping structure $V_0$ is evaluated by the revised FG-$k$-means on input dataset. The objective function of revised FG-$k$-means is defined as

$$
P(U, Z, V, W) = \sum_{l=1}^{k} [\sum_{i=1}^{n} \sum_{t=1}^{T} \sum_{j \in G_t} h_{i,l} w_{l,t} v_{l,j} d(x_{i,j}, z_{l,j}) + \lambda \sum_{t=1}^{T} w_{l,t} \log(w_{l,t})] \tag{6}
$$

subject to

$$
\begin{cases} \sum_{l=1}^{k} h_{i,l} = 1, \ i = 1, 2, \cdots, n \\ \sum_{l=1}^{k} w_l = 1 \\ V_l V_l^T = I \end{cases}, \tag{7}
$$

where $X = \{x_i \mid x_i \in \mathbb{R}^m\}_{i=1}^{n}$ is the set of $n$ objects with $m$ features, $H = \{h_{i,l} \mid h_{i,l} \in \{0,1\}\}$ is the set of membership indicators of $n$ objects in $k$ clusters, $Z = \{z_l \mid z_l \in \mathbb{R}^m\}_{l=1}^{k}$ is the set of $k$ cluster centers, and $V_l$ is specified based on $V_0$, i.e., $V_l = V_0 \circ V_l$.

There are two revisions made on the original FG-*k*-means [7]. The mass-based dissimilarity is used to replace the Euclidean distance and the orthogonal constraint is set on the matrices of individual feature weights $V_l$. Below, we discuss the techniques to solve the aforementioned optimization problem.

### 4.2. The Mass-Based Dissimilarity

Due to the dimensionality curse in the high-dimensional space, the Euclidean distances between objects tend to be similar. To overcome this problem, we choose to use the mass-based dissimilarity studied in [27]. This dissimilarity measures the mass dissimilarity depending on data distribution, i.e., the probability mass of the smallest region covering the two objects. It provides a more effective match than the distance measures in the nearest neighbor search for *k*-nn classifiers and information retrieval. It is named $m_p$-dissimilarity and defined in the same form as $l_p$-norm, except that the dissimilarity in dimension $i$ is the probability mass in a region $P(R_i(x; y))$ rather than the distance $\|x_i - y_i\|$.

Let $D$ be the data domain with the probability density $F$. The mass-based dissimilarity of objects $x$ and $y$ in $D$ is defined as $P(R(x, y | H; D))$, where $H \in H(D)$ is a hierarchical model to partition the data space into non-overlapping and non-empty regions. The smallest local region $R(x, y | H; D)$ covering $x$ and $y$ with regard to $H$ and $D$ is defined as

$$R(x, y | H; D) = \arg \min_{\{x, y\} \in r} \sum_{l \in D} \mathbf{1}(l \in r),\tag{8}$$

where $\mathbf{1}(\cdot)$ is an indicator function.

The mass-based dissimilarity measure of $x$ and $y$ with regard to $H$ and $D$ is defined as the probability of $R(x, y | H; D)$, i.e.,

$$d(x, y) = E_{\mathcal{H}(D)} [P_F R(x, y | H; D)],\tag{9}$$

where $P_F(i)$ is the probability with regard to $F$ and the expectation is taken over all models in $\mathcal{H}(D)$. In practice, the mass-based dissimilarity is estimated from a finite number of models $H_i \in \mathcal{H}(D)$, $i = 1, 2, \cdots, t$ as

$$d(x, y) = \frac{1}{t} \sum_{i=1}^{t} \widetilde{P}(R(x, y | H_i; D)),\tag{10}$$

where $\widetilde{P}(R) = \frac{1}{|D|} \sum_{z \in D} \mathbf{1}(z) \in R$. Note that $d(x, y)$ is the smallest local region covering $x$ and $y$ and it is analogous to the shortest distance between $x$ and $y$ used in the geometric model.

### 4.3. Optimization of Equation (6)

We minimize the objective function in Equation (6) by iteratively solving the following four minimization problems:

1. Problem $P_1$: Fix $Z = \hat{Z}$, $V = \hat{V}$ and $W = \hat{W}$, and solve the reduced problem $P(U)$. Problem $P_1$ is solved by

$$\begin{cases} u_{i,l} = 1, & \text{if } D_l \leq D_s \text{ for } 1 \leq s \leq k \\ u_{i,s} = 0, & \text{otherwise} \end{cases},\tag{11}$$

   where $D_s = \sum_{t=1}^{T} w_{s,t} \sum_{j \in G_t} v_{s,j} d(x_{i,j}, z_{s,j})$.

2. Problem $P_2$: Fix $U = \hat{U}$, $V = \hat{V}$ and $W = \hat{W}$, and solve the reduced problem $P(Z)$. Problem $P_2$ is solved by updating the centers of the clusters by Algorithm 1.

---

**Algorithm 1** Cluster center updating.

---

**Input:** $U, W, V$.

**Output:** The updating cluster centers $Z$.

1: Generate a similarity matrix $S \in \mathbb{R}^{n*n}$, where $s_{i,j} = s_{j,i} = u_{i,j} d(x_i, x_j)$;

2: Generate a distance sum matrix $S_{sum} \in \mathbb{R}^{1*n}$, where $s_i = \sum_{j=1}^{n} s_{j,i}$;

3: Generate a density matrix $D \in \mathbb{R}^{1*n}$, find the smallest value in every column $m$ of $D * U$ and

consider the corresponding object as the center of cluster $m$ for $m = 1, 2, \cdots, k$.

4: **return** $Z$;

---

3.  Problem $P_3$: Fix $U = \hat{U}$, $Z = \hat{Z}$, and $V = \hat{V}$, and solve the reduced problem $P(W)$. Problem $P_3$ is solved by the following Theorem 1.

**Theorem 1.** *Let $U = \hat{U}$, $Z = \hat{Z}$, and $V = \hat{V}$ be fixed and $\lambda > 0$, $P(W)$ is minimized iff*

$$w_{l,t} = \exp \frac{-E_{l,j}}{\lambda} / \sum_{j \in G_t} \exp \frac{-E_{l,j}}{\lambda}, \tag{12}$$

*where*

$$D_{l,t} = \sum_{i=1}^{n} \widehat{u_{i,l}} \widehat{v_{l,j}} d(x_{i,j}, \widehat{z_{l,j}}). \tag{13}$$

**Proof.** Given $U = \hat{U}$, $Z = \hat{Z}$ and $V = \hat{V}$, we minimize the objective function with respect to $W$. Since there exists a set of $k \times T$ constraints $\sum_{t=1}^{T} w_{l,t} = 1$, we form the Lagrangian by isolating the terms which contain $\{w_{l,1}, w_{l,2}, \cdots, w_{l,t}\}$ and adding the appropriate Lagrangian multipliers as

$$L_{\{w_{l,1}, w_{l,2}, \cdots, w_{l,T}\}} = \sum_{t=1}^{T} [w_{l,t} D_{l,t} + \lambda \sum_{t=1}^{T} \sum_{j \in G_t} w_{l,t} \log w_{l,t} v_{l,j} \log v_{l,j} + \gamma_{l,t} (\sum_{j \in G_t} w_{l,t} - 1)], \tag{14}$$

where $D_{l,t}$ is a constant in the $t$th feature group on the $l$th cluster for fixed $U = \hat{U}$, $Z = \hat{Z}$ and $V = \hat{V}$. By setting the gradient of $L_{\{w_{l,1}, w_{l,2}, \cdots, w_{l,T}\}}$ with respect to $\gamma$ and $w_{l,t}$ to zero, we obtain

$$\frac{\partial L_{\{w_{l,1}, w_{l,2}, \cdots, w_{l,T}\}}}{\partial \gamma} = \sum_{t=1}^{T} w_{l,t} - 1 = 0 \tag{15}$$

and

$$\frac{\partial L_{\{w_{l,1}, w_{l,2}, \cdots, w_{l,T}\}}}{\partial w_{l,t}} = D_{l,t} + \lambda \sum_{j \in G_t} v_{l,j} \log v_{l,j} (1 + \log w_{l,t}) + \gamma = 0, \tag{16}$$

where $t$ is the index of feature group to which the $j$th feature is assigned. Then, we obtain

$$w_{l,t} = \exp \frac{-D_{l,t}}{\lambda} / \sum_{t=1}^{T} \exp \frac{-D_{l,t}}{\lambda}. \tag{17}$$

$\square$

4.  Problem $P_4$: Fix $U = \hat{U}$, $Z = \hat{Z}$, and $W = \hat{W}$, and solve the reduced problem $P(V)$. Problem $P_4$ is solved as follows. Because of the additivity of objective function in Equation (6), the matrix $W$ can be divided into $k$ subproblems for $k$ clusters, respectively. Let

$$Q_l = diag(w_l)^T diag(w_l) \tag{18}$$

and

$$q_{i,l} = h_{i,l} \left( x_i - z_l \right), \tag{19}$$

then the *l*th subproblem of original problem can be written as

$$\min_{V \in \mathbb{R}_+^{t \times m}} \quad \sum_{i=1}^{n} q_{i,l}^T V_l^T Q_l V_l q_{i,l}. \tag{20}$$
$$\text{s.t. } V_l V_l^T = I$$

The subproblem in Equation (20) has the nonnegative and orthogonal constraints on matrix $V_l$, which makes the problem NP-hard to solve directly. The methods used here are analogous to that of nonnegative matrix factorization (NMF). We replace the orthogonal constraint with a *F*-norm measurement of orthogonality as the relaxation, i.e.,

$$\min_{V \in \mathbb{R}_+^{t \times m}} \quad \sum_{i=1}^{n} q_{i,l}^T V^T Q_l V q_{i,l} + \frac{\eta}{2} (VV^T - I)_F^2, \tag{21}$$

where $\eta \geq 0$ is a parameter to control the orthogonality of $V$ explicitly. The Lagrangian of Equation (21) is

$$L(V, \Lambda) = \sum_{i=1}^{n} q_{i,l}^T V^T Q_l V q_{i,l} + \frac{\eta}{2} (VV^T - I)_F^2 - tr(\Lambda V^T), \tag{22}$$

where $\Lambda$ is the Lagrange multiplier for constraint $V \in \mathbb{R}_+^{t \times m}$. By $\nabla_V L = 0$, we have

$$2QVSS^T + 2\eta (VV^T - I)V - \Lambda = 0, \tag{23}$$

where $S = [q_{1,l}, q_{2,l}, \ldots, q_{n,l}] \in \mathbb{R}^{m \times n}$. According to the KKT complementary condition on $[V]_{i,j} \geq 0$, by making a Hadamard product with $V$ on both sides of Equation (23), we obtain

$$(QVSS^T + \eta (VV^T - I)V) \circ V = 0. \tag{24}$$

The multiplicative updating rule for $V_l$ is derived as

$$V_{i,j} \leftarrow V_{i,j} \frac{[QV(SS^T)^- + \eta V]_{i,j}}{[QV(SS^T)^+ + \eta VV^T V]_{i,j}}, \tag{25}$$

where $\eta$ is a parameter to control the orthogonality among different rows of $V$, and $()^+$ and $()^-$ are the operators to get the positive and negative parts of the input matrix, respectively, i.e.,

$$[(A)^+]_{i,j} = \begin{cases} [A]_{i,j} & \text{if } [A]_{i,j} > 0 \\ 0 & \text{otherwise} \end{cases} \tag{26}$$

and

$$[(A)^-]_{i,j} = \begin{cases} |[A]_{i,j}| & \text{if } [A]_{i,j} < 0 \\ 0 & \text{otherwise} \end{cases}. \tag{27}$$

In the next section, we introduce the method to optimize the latent feature grouping structure $V_0$.

### 4.4. Evolutionary Method to Select the Best Feature Grouping Structure $V_0$

The Darwinian evolutionary process [28] is used to search the best feature grouping structure $V_0$. In this process, the feature grouping structures $V_0$ are encoded as the chromosomes and the revised FG-*k*-means is used as the fitness function to evaluate the chromosomes. The best $V_0$ is selected through

the evolutions of generations. To our knowledge, this is the first attempt to use the evolutionary process to search the best feature grouping structure from high-dimensional data.

We first present a classical evolutionary method where the population looks for the best grouping from the feature set. Each individual chromosome encodes a feature grouping structure $V_0$. The chromosome $A^{i,g}$ of the $i$th individual in the $g$th generation is defined as

$$A^{i,g} = (V_1^{i,g}, \cdots, V_k^{i,g}, \cdots, V_m^{i,g}), \tag{28}$$

where $A^{i,g}$ is a binary sequence with length $t$, $V_k^{i,g}$ is the $k$th column of matrix $V_0$ and $m$ is the number of features in the dataset. For example, the matrix $V_0$ in Equation (3) is encoded as

$$A^{i,g} = \{(1,0,0), (1,0,0), (0,0,1), (0,1,0), (0,0,1), (0,1,0)\}. \tag{29}$$

We can see that each binary sequence $V_k^{i,g}$ has only one element as 1 and the rest as 0. This is a constraint on the structure of chromosomes. To start the evolutionary process, there are 20 chromosomes that are generated randomly as the first generation of individuals. To generate the binary sequences for each chromosome, one position is randomly selected from $t$ possible positions and is set as value 1. The remaining $t-1$ positions are set as 0.

After all chromosomes are initialized, they are evaluated by the revised FG-$k$-means algorithm. From each chromosome, the matrix $V_0$ is constructed. The matrix $V_l$ is initialized by solving $V_l V_l^T = I$. Since the solutions are not unique, the different initial $V_l$s for different clusters are initialized. Then, the initial $V_l$s are obtained by $V_l = V_0 \circ V_l$.

The initial feature group weights and initial cluster centers are generated and selected randomly. The number of clusters $k$ is given. The revised FG-$k$-means algorithm is executed on the input dataset once for each chromosome to produce one clustering result. The Bayesian information criterion (BIC) is used to evaluate the clustering result and score the chromosome. After all chromosomes are scored, the genetic operations such as selection, crossover and mutation are applied to the chromosomes to produce the new individual chromosomes for next generation as follows.

- There are 10 strongest chromosomes which are selected with the highest scores.
- The crossover is performed in the following steps. The 10 chromosomes are randomly grouped into five pairs. For each pair of chromosome $i$ and chromosome $j$, the corresponding binary sequence $V_k^{i,g}$ and $V_k^{j,g}$ are compared. If two sequences are same, the sequence is copied as the new generation of $V_k^{s,g+1}$. For the remaining pairs of different binary sequences, we randomly select one sequence from one chromosome to replace the corresponding sequence of another chromosome by the probability $\alpha_k \in [0,1]$. Finally, we encode $V_0$ as a new chromosome in the next generation. The rule to generate $V_k^{s,g+1}$ is defined as

$$V_k^{s,g+1} = \begin{cases} V_k^{i,g} & \text{if } V_k^{i,g} = V_k^{j,g} \text{ or } \alpha_k \geq 0.5 \\ V_k^{j,g} & \text{otherwise} \end{cases}, \tag{30}$$

where $\alpha_k$ is randomly generated for each $V_k^{s,g+1}$.

- For the process of mutation, we randomly choose five chromosomes from 10 alternative chromosomes. For each chromosome $A^{i,g}$, we randomly generate a new chromosome $A_k^{rand} = (V_1^{rand}, \cdots, V_k^{rand}, \cdots, V_m^{rand})$. The rule to generate $V_k^{i,g+1}$ is

$$V_k^{i,g+1} = \begin{cases} V_k^{i,g} & \text{if } \alpha_k \geq 0.5 \\ V_k^{rand} & \text{otherwise} \end{cases}, \tag{31}$$

where $\alpha_k$ is randomly generated for each $V_k^{i,g+1}$.

In this way, we generate 10 new chromosomes and combine them with the 10 strongest chromosomes to form a new population for exploration and exploitation in the next generation of evolution.

*4.5. LFGL Algorithm*

The process of learning the latent feature grouping structure $V_0$ consists of three stages, i.e., the individual feature weights, the feature group weights and a chromosome score from the input dataset. The initialization stage generates the first generation of 20 chromosomes representing 20 initial $V_0$s. The second stage uses the revised FG-$k$-means algorithm to score the 20 chromosomes. The third stage selects the 10 strongest chromosomes according to the scores and performs the genetic operations on the selected chromosomes to produce the new generation of chromosomes for evolution. This process continues until the termination criterion is met. The evolution process of LFGL algorithm is summarized in Algorithm 2.

---

**Algorithm 2** LFGL algorithm.

---

**Input:** The dataset $X$, the number of clusters $k$, two positive parameters $\lambda$ and $\eta$, the number of feature groups $t$.

**Output:** Local optimal values of $\mathcal{H}$, $\mathcal{Z}$, $\mathcal{V}$, and $\mathcal{W}$.

1: Initialize 20 chromosomes representing 20 different possibilities of feature grouping;
2: For each chromosome, we initialize $\mathcal{W}$ by sampling the positive values $[w_l]_i \sim \mathcal{N}(1, 0.01)$, then normalize $w_l$ so that $1^T w_l = 1$;
3: Initialize $\mathcal{V}$ with the method mentioned in Section 4.4 to build $V$ matrix, then normalize $V_l$ so that the $\ell^2$-norm of each row $V_l$ of is 1;
4: Randomly choose $k$ cluster centers $Z^0$;
5: Update $H^{t+1}$, $Z^{t+1}$, $W^{t+1}$ and $V^{t+1}$, respectively;
6: The objective function $P$ obtains its local minimum value, then update $V^{t+1}$ and go back to Step 9;
7: Calculate BIC of 20 clustering results from 20 chromosomes, choose the best 10 ones and make 10 new chromosomes by crossover and mutation;
8: Repeat ten times and find the best solution of clustering.

---

## 5. Experiments

We tested the performance of LFGL algorithm on several datasets in high dimensions. The results were compared with five existing clustering algorithms: $k$-means [29,30], TWKM [6], EWKM [4], LAC [16] and FG-$k$-means [7].

*5.1. Datasets*

Seven high-dimensional datasets in the real world were used to evaluate LFGL algorithm, where six genetic datasets were downloaded from http://archive.ics.uci.edu/ml/datasets.html and one text dataset was obtained from http://www.escience.cn/people/fpnie/papers.html. The common characteristics of these datasets are small numbers of objects with large numbers of features. The details of the datasets are listed in Table 1.

**Table 1.** Dataset Description.

| Dataset | Objects | Features | Classes |
|---|---|---|---|
| SRBCT | 63 | 2308 | 4 |
| Lymphoma | 62 | 4026 | 3 |
| Prostate | 102 | 6033 | 2 |
| Adenocarcinoma | 76 | 9868 | 2 |
| Breast2classes | 77 | 4870 | 2 |
| CNS | 60 | 7129 | 2 |
| WebKB texas | 814 | 4029 | 7 |

*5.2. Evaluation Measures*

Five measures were used to evaluate the clustering results of six algorithms. Each algorithm was run on each dataset 100 times to produce 100 results. The average value of 100 results on each dataset corresponding to each algorithm was used as the measure of algorithm. Let $L$ be the partition of a dataset by the labeled classes and $\widehat{L}$ the partition of the clustering result by algorithms. The confusion matrix can be generated, as shown in Table 2, to calculate the correspondence between the true clusters and results. The elements of $TP$, $FN$, $FP$ and $TN$ are the numbers of objects satisfying both true and predicted conditions. The five measures are defined based on the confusion matrix as follows.

- **Accuracy** is defined as

$$Accuracy = \frac{TP + TN}{TP + FP + TN + FN}; \tag{32}$$

- **Rand Index** is defined as

$$Rand\ Index = \frac{TP + FN}{TP + FP + TN + FN}; \tag{33}$$

- **Precision** is defined as

$$Precision = \frac{TP}{TP + FP}; \tag{34}$$

- **Recall** is defined as

$$Recall = \frac{TP}{TP + FN}; \tag{35}$$

- **F-measure** is defined as

$$F\text{-}measure = \frac{2 * Precision * Recall}{Precision + Recall}. \tag{36}$$

**Table 2.** Confusion matrix.

| Data | Predicted Positive | Predicted Negative |
|---|---|---|
| True condition positive | True Positive ($TP$) | False Negative ($FN$) |
| True condition negative | False Positive ($FP$) | True Negative ($TN$) |

*5.3. Parameters Settings*

In Algorithm 2, we include two parameters $\lambda$ and $\eta$ which may impact the performance of LFGL. We set the parameter $\lambda$ as $\{1, 2, 3, 4, 5, 8, 10, 14, 16, 20\}$ and $\eta$ as $\{1, 2, 3, 4, 5, 6, 7, 8, 9, 10\}$. For each combination of $\lambda$ and $\eta$, we ran LFGL 100 times and recorded the five measurements mentioned above. We ran the tests on dataset Prostate. The results are illustrated in Figure 2. We did not observe significant rule of the parameters. Hence, $\lambda = 1$ and $\eta = 1$ were determined for the followed experiments.

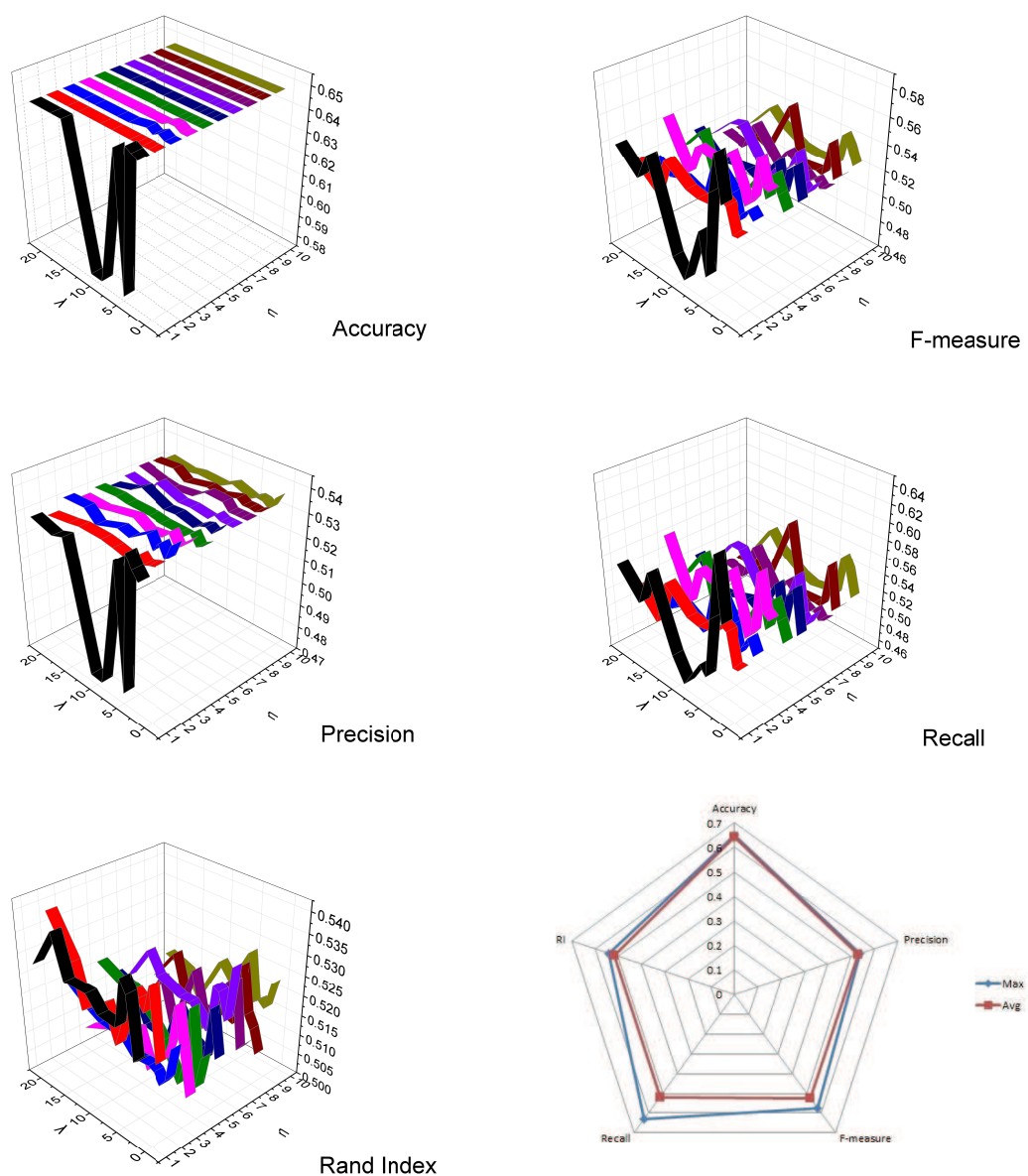

**Figure 2.** The parameter control for Prostate dataset.

## 5.4. Clustering Results and Analysis

The clustering results on six genetic datasets and the text dataset are shown in Table 3. From the results, we can see that LFGL algorithm outperformed all five other clustering algorithms on most datasets. If we consider all clustering results, LFGL algorithm significantly outperformed all other five clustering algorithms on Prostate dataset. On other datasets, LFGL algorithm produced similar results as the five other clustering algorithms. These results show that LFGL algorithm is effective in clustering high-dimensional data. LFGL algorithm is established particularly with a target on the Genetic datasets to investigate the relations between human genes and diseases. It is an extra gain that it also performed well on the text dataset.

**Table 3.** Summary of clustering results corresponding to six clustering algorithms.

| Data | Evaluation | *k*-Means | EWKM | TWKM | LAC | FG-*k*-Means | LFGL |
|---|---|---|---|---|---|---|---|
| SRBCT | Rand Index | 0.661 | 0.603 | 0.654 | 0.597 | 0.654 | 0.777 |
| | Accuracy | 0.528 | 0.476 | 0.530 | 0.413 | 0.555 | 0.512 |
| | Precision | 0.376 | 0.304 | 0.366 | 0.296 | 0.375 | 0.351 |
| | Recall | 0.353 | 0.344 | 0.380 | 0.339 | 0.385 | 0.428 |
| | F-Measure | 0.293 | 0.163 | 0.359 | 0.293 | 0.287 | 0.339 |
| Lymphoma | Rand Index | 0.727 | 0.804 | 0.612 | 0.488 | 0.921 | 0.929 |
| | Accuracy | 0.855 | 0.838 | 0.755 | 0.419 | 0.951 | 0.845 |
| | Precision | 0.844 | 0.854 | 0.677 | 0.488 | 0.966 | 0.663 |
| | Recall | 0.563 | 0.736 | 0.441 | 0.336 | 0.857 | 0.926 |
| | F-Measure | 0.675 | 0.791 | 0.534 | 0.488 | 0.918 | 0.771 |
| Prostate | Rand Index | 0.507 | 0.504 | 0.507 | 0.495 | 0.509 | 0.524 |
| | Accuracy | 0.578 | 0.606 | 0.576 | 0.500 | 0.578 | 0.593 |
| | Precision | 0.503 | 0.49 | 0.502 | 0.491 | 0.500 | 0.507 |
| | Recall | 0.541 | 0.515 | 0.573 | 0.510 | 0.523 | 0.671 |
| | F-Measure | 0.521 | 0.507 | 0.532 | 0.491 | 0.511 | 0.579 |
| Adenocarcinoma | Rand Index | 0.528 | 0.730 | 0.605 | 0.588 | 0.552 | 0.730 |
| | Accuracy | 0.842 | 0.842 | 0.832 | 0.615 | 0.842 | 0.843 |
| | Precision | 0.722 | 0.743 | 0.731 | 0.642 | 0.708 | 0.738 |
| | Recall | 0.576 | 0.964 | 0.723 | 0.562 | 0.661 | 0.807 |
| | F-Measure | 0.641 | 0.839 | 0.719 | 0.692 | 0.682 | 0.768 |
| Breast2classes | Rand Index | 0.546 | 0.507 | 0.479 | 0.491 | 0.508 | 0.547 |
| | Accuracy | 0.662 | 0.584 | 0.574 | 0.470 | 0.584 | 0.611 |
| | Precision | 0.537 | 0.507 | 0.452 | 0.404 | 0.506 | 0.523 |
| | Recall | 0.696 | 0.757 | 0.608 | 0.603 | 0.682 | 0.763 |
| | F-Measure | 0.607 | 0.608 | 0.527 | 0.481 | 0.599 | 0.611 |
| CNS | Rand Index | 0.493 | 0.506 | 0.496 | 0.500 | 0.512 | 0.544 |
| | Accuracy | 0.661 | 0.661 | 0.661 | 0.615 | 0.555 | 0.595 |
| | Precision | 0.531 | 0.542 | 0.536 | 0.507 | 0.540 | 0.485 |
| | Recall | 0.583 | 0.584 | 0.545 | 0.609 | 0.683 | 0.737 |
| | F-Measure | 0.556 | 0.559 | 0.539 | 0.513 | 0.540 | 0.485 |
| WebKB texas | Rand Index | 0.507 | 0.507 | 0.496 | 0.488 | 0.501 | 0.523 |
| | Accuracy | 0.578 | 0.563 | 0.516 | 0.528 | 0.545 | 0.593 |
| | Precision | 0.502 | 0.502 | 0.492 | 0.509 | 0.496 | 0.511 |
| | Recall | 0.757 | 0.514 | 0.557 | 0.562 | 0.564 | 0.612 |
| | F-Measure | 0.603 | 0.508 | 0.520 | 0.513 | 0.496 | 0.507 |

*5.5. Feature Grouping Analysis*

We conducted the experiments to investigate the trend of evolution of chromosomes, i.e., the feature grouping structure $V_0$ in the evolutionary process. Figure 3 shows the results on the dataset SRBCT. Figure 3a shows the Rand Index measures of 10 strongest chromosomes in different generations. We can see that the clustering results of LFGL algorithm improved with the increase of generations in the evolutionary process and became stable after some generations. This indicates that the evolutionary process optimized the clustering through searching the optimal feature grouping structure $V_0$. Figure 3b shows the standard deviations of mutual dissimilarities between 10 strongest chromosomes in each generation. We can see the continuous dropping of standard deviation with the increase of generations. This indicates that the strongest chromosomes tended to become similar during the evolution, which implies the convergence of optimal feature grouping structure.

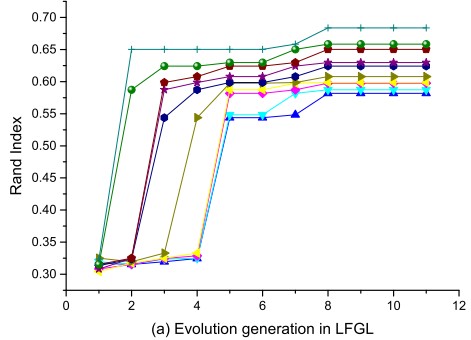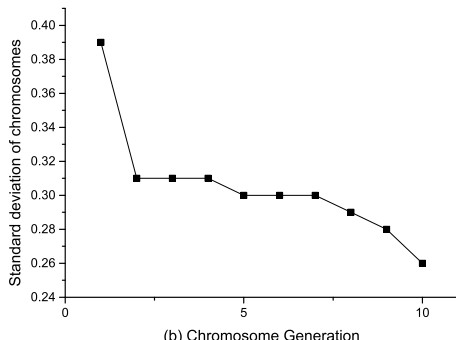

(a) Evolution generation in LFGL
(b) Chromosome Generation

**Figure 3.** The comparison of chromosomes during the evolution process.

## 6. Conclusions and Future Works

In this paper, we present a new method to automatically find the latent feature grouping structure in high-dimensional data. The latent feature group learning (LFGL) algorithm is proposed to cluster high-dimensional data from subspaces of feature groups and individual features. The Darwinian evolution process is used to search the optimal group structures. The revised FG-*k*-means is used to evaluate the feature grouping and cluster the data. The experimental results on different kinds of datasets show that LFGL algorithm outperformed five existing clustering algorithms. Meanwhile, the results of clustering were evaluated for the accuracy of feature groupings. The future works will mainly focus on two directions. First, we will seek real applications for LFGL algorithm. The integration of LFGL algorithm with feature selection [31,32] to improve the generalization of learning algorithm will be very promising future work. Second, we will extend LFGL algorithm to big data analysis and management [33,34].

**Author Contributions:** Methodology, W.W.; Conceptualization, Y.H.; Validation, L.M.; Writing-Original Draft Preparation, W.W.; Writing-Review & Editing, Y.H.; Supervision, J.Z.H.

**Funding:** We would like to thank the editors and two anonymous reviewers whose meticulous readings and valuable suggestions helped us to improve this paper significantly. This paper was supported by National Key R&D Program of China (2017YFC0822604-2), China Postdoctoral Science Foundation (2016T90799) and Scientific Research Foundation of Shenzhen University for Newly-introduced Teachers (2018060).

**Conflicts of Interest:** The authors declare no conflict of interest.

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
