# Peer review of "Latent Feature Group Learning for High-Dimensional Data Clustering"

_information, doi:10.3390/info10060208_

Reviewer 1 Report

This paper address the problem of identifying feature grouping structures in high-dimensional data using an unsupervised approach. The paper is very interesting and I think it could be published after some improvements.

- The experiments are performed in classification datasets. So, I would recommend to include two supervised algorithms [1,2] in the comparison. This would increase the quality of the work.

- I would recommend to include supervised works in the related work.

[1] Song, Q., Ni, J., Wang, G., 2013. A fast clustering-based feature 897 subset selection algorithm for high-dimensional data. IEEE Transactions on Knowledge and Data Engineering (1), 1–14.

[2] García-Torres, M., Gómez-Vela, F., Melián-Batista, B., Moreno-Vega, J.M., 2016. High-dimensional feature selection via feature grouping. Information Sciences (326), 102-118.

Author Response

Authors‘ response to the comments of Reviewer #2 on information-486486:

Latent feature group learning for high-dimensional data clustering

Authored by Wen-Ting Wang, Yu-Lin He, Li-Heng Ma, Joshua Zhexue Huang

I thank the anonymous Reviewer #1 very much for his/her evaluation and suggestion on our paper.

Reviewer #1:

This paper address the problem of identifying feature grouping structures in high-dimensional data using an unsupervised approach. The paper is very interesting and I think it could be published after some improvements.

Response: Thank Reviewer #1 very much for your evaluation and suggestion on information-486486.

We have seriously considered your suggestion and improved the manuscript accordingly.

The experiments are performed in classification datasets. So, I would recommend to include two supervised algorithms [1,2] in the comparison. This would increase the quality of the work. I would recommend to include supervised works in the related work.

[1] Song, Q., Ni, J., Wang, G., 2013. A fast clustering-based feature subset selection algorithm for high-dimensional data. IEEE Transactions on Knowledge and Data Engineering (1), 1–14.

[2] García-Torres, M., Gómez-Vela, F., Melián-Batista, B., Moreno-Vega, J.M., 2016. High-dimensional feature selection via feature grouping. Information Sciences (326), 102-118.

Response: Thanks a lot for your enlightenment.

Yes, the integration of LFGL algorithm with feature selection to improve the generalization of learning algorithm will be a very promising future work.

We carefully studied the recommended works which have been included in the reference-list (please check Refs. [31] and [32]) of our submission and hope that some exciting research results will be achieved in the future.

Reviewer 2 Report

This paper proposes a new method of clustering, as long as feature grouping. Several mathematical manipulations are used to enhance the accuracy of the clustering compared to the previous algorithms. The innovation of the paper is significant when dealing with the ultra-high dimensional data sets from different areas of real life. 

Author Response

Authors‘ response to the comments of Reviewer #2 on information-486486:

Latent feature group learning for high-dimensional data clustering

Authored by Wen-Ting Wang, Yu-Lin He, Li-Heng Ma, Joshua Zhexue Huang

I thank the anonymous Reviewer #2 very much for his/her evaluation on our paper.

Reviewer #2:

This paper proposes a new method of clustering, as long as feature grouping. Several mathematical manipulations are used to enhance the accuracy of the clustering compared to the previous algorithms. The innovation of the paper is significant when dealing with the ultra-high dimensional data sets from different areas of real life.

Response: Thank Reviewer #2 very much for your evaluation on information-486486.
